# Error Overboundings of KF-Based IMU/GNSS Integrated System Against IMU Faults

**DOI:** 10.3390/s19224912

**Published:** 2019-11-11

**Authors:** Wei Liu, Dan Song, Zhipeng Wang, Kun Fang

**Affiliations:** School of Electronic and Information Engineering, Beihang University, Beijing 100191, China; lw1014hjh@163.com (W.L.); songdan0207@buaa.edu.cn (D.S.); fangkun@buaa.edu.cn (K.F.)

**Keywords:** error overbounding, fault propagation, inertial measurement unit (IMU)/global navigation satellite system (GNSS) integration, integrity, unmanned aerial vehicle (UAV) navigation

## Abstract

Considering the inertial measurement unit (IMU) faults risk of an unmanned aerial vehicle (UAV), this paper studies the error overboundings of the state estimation of the extended Kalman filter (EKF) in a tightly coupled IMU/global navigation satellite system (GNSS) integrated architecture under the IMU fault condition, which can be used to assure the integrity of the UAV navigation system. The error overboundings of the error-state inertial navigation equations based EKF (error-state EKF) are obtained according to the IMU faults propagation derivation, which can be expressed as a sum of the terms related to the EKF innovation, the estimated bias, and the remaining position error. It presents the same expression with the error overbounding of the full-state inertial navigation equations based EKF (full-state EKF). Simulation results show that both the error overboundings of the error-state and full-state EKFs can fit the state error against the IMU faults, but the error-state EKF is more suitable for UAV navigation system integrity assurance due to its higher calculation efficiency. This study will be extended to the integrity monitoring of multisensor systems.

## 1. Introduction

Unmanned aerial vehicles (UAVs) are widely used in many aspects of our daily life. The application of UAVs will permeate more aspects of public life in the near future. To ensure the safety of a UAV, its navigation solution should be fully protected by an integrity assurance mechanism, including real-time fault detection and an integrity risk evaluation. Integrity risk is the probability of hazardous misleading information (HMI) [1] and can be evaluated by the error overboundings (i.e., protection levels (PLs)) of the estimated position solution corresponding to the required probability of HMI, i.e., the integrity risk requirement [2,3].

An UAV usually adopts an inertial measurement unit (IMU)/global navigation satellite system (GNSS) integrated navigation system. The configuration of the IMU on UAVs can be performed according to the practical application requirements. UAVs similar to the DJI Phantom series use two IMUs to provide redundant measurements [4]. The experimental test carried out by Rhudy uses four IMUs to provide redundant IMU data [5]. Although the redundantly configured IMU can be used to detect IMU faults, there is still a probability of missed detection, and the fault risk of IMU still exists. Moreover, the low-cost micro-electromechanical system (MEMS) IMU on an UAV has a lower reliability and is more vulnerable to the flight environment than the fiber-optic strapdown IMU on civil aviation aircraft. Giorgio showed that IMUs are more prone to malfunction in high-vibration environments, where UAVs typically operate [6]. Bhatti listed the fault types of IMU sensors and summarized the potential causes of each fault type [7]. The IMU also has a high fault risk in the integrated navigation system for UAVs [8]. Therefore, IMU faults cannot be ignored in the integrity assurance mechanism of UAVs’ integrated navigation architecture.

Many studies have been carried out to assure the navigation integrity of UAVs. However, most studies still place emphasis on assuring the integrity of the GNSS. Some of the studies focus on researching GNSS integrity augmentation systems, which apply GNSS monitoring technology in civil aviation to UAVs. For instance, Pullen developed a local-area differential GNSS architecture around the concept of local-area UAV network operations, which can assure the integrity of UAVs against GNSS faults with a low cost [9]. Lee proposed a high-integrity navigation architecture for the integrated IMU/LAD-GNSS system with a filter based on the extended Kalman filter (EKF), which can support continuous and reliable navigation by overcoming the GNSS’ signal vulnerability [10]. Kim presented a LAD-GNSS architecture for UAVs that communicates with ground subsystems in real time, reducing the vertical PLs by approximately 16% [11].

Other studies have utilized the EKF innovation or residual to monitor the GNSS integrity, which is known as IMU-aided GNSS integrity monitoring. There are two aspects of this research—IMU-aided GNSS fault detection and integrity risk evaluation against a GNSS fault. For the first aspect, Brenner proposed the multiple solution separation (MSS) method [12], Diesel investigated the autonomous integrity monitoring by extrapolation (AIME) method [13], and Crespillo performed a comparison between innovation-based and residual-based fault detection in the EKF for GNSS/INS hybridization [14]. They are all effective for GNSS fault detection in an IMU/GNSS navigation system. For the second aspect, Joerger evaluated the integrity risk in a sequential Kalman filter (KF) by quantifying the worst-case integrity risk based on the relationship between the state error and the GNSS range residual to construct monitor test statistics using a least square (LS) expression [15]. Tanil instead used the recursive expression and applied it to GNSS spoofing detection [16]. Bhattacharyya proposed the real-time PLs computation through the residual-based RAIM algorithm [17]. All of these approaches can monitor the integrity of the GNSS in the IMU/GNSS integrated navigation system.

Different from the above studies, Lee proposed an integrity assurance mechanism for an EKF-based IMU/GNSS integrated system against IMU faults. He calculated the PLs using the EKF innovations and additional uncertain noise boundary terms, by deriving the IMU faults propagation process in the EKF [18]. In his study, the full-state inertial navigation equations, including the attitude, velocity, and position update equations, were used as the EKF system propagation equations. However, only the velocity update equation was used as an example to explain how the IMU faults propagate in the EKF. Since the inertial navigation equation contains three update equations of the position, velocity, and attitude, and the faults have different expressions in the three update equations. The attitude and position update equations also need to be considered for analyzing the IMU faults propagation process.

In our study, the error-state inertial navigation equations, including the attitude error, the velocity error, and the position error update equations, were used to analyze the IMU faults propagation process which are more commonly used as the EKF system propagation equations in IMU/GNSS integrated navigation than the full-state inertial navigation equations [19,20]. The error overboundings against the IMU faults for the EKF state estimation using the error-state inertial navigation equation is derived in this paper. Moreover, the error overboundings using error-state and full-state inertial navigation equations are compared to determine which kind is more conducive to assuring the integrity for the IMU/GNSS integrated navigation architecture against IMU faults. To simplify the description below, the EKF using the full-state inertial navigation equations is named the full-state EKF, whereas the EKF using the error-state inertial navigation equation is named the error-state EKF.

Accordingly, the remainder of this paper is organized as follows: (1) the general structures of the IMU/GNSS integrated navigation architecture using the IMU error-state model are introduced, (2) the detailed IMU faults propagation process in the error-state EKF is specially derived, (3) a description of the real-time error overboundings (PLs) equations in the error-state EKF for IMU faults based on innovations, (4) the simulations for the error overboundings calculation under IMU faults are presented, and (5) the summary and concluding remarks close the paper.

## 2. Architecture of the IMU/GNSS Integrated Navigation System

### 2.1. IMU/GNSS Integration Architecture

Figure 1 illustrates an overview of the EKF-based tightly coupled IMU/GNSS integration architecture. The inputs are the GNSS pseudo-range, ρ, pseudo-range rate, $\dot{\rho }$, GNSS receiver clock offset, $\delta \rho_{c}$, and GNSS receiver drift, $\delta {\dot{\rho }}_{c}$, from the GNSS ranging processor and the IMU position, $P_{IMU}$, velocity, $V_{IMU}$, and attitude, $A_{IMU}$, navigation parameters from the inertial navigation processor. The outputs of the integrated navigation system are the corrected IMU position, $\hat{P}$, velocity, $\hat{V}$, and attitude, $\hat{A}$. The integration is completed in the range domain. In this study, the error-state inertial navigation equations based EKF (i.e., the error-state EKF) was used for the IMU/GNSS integration architecture, and the outputs of the IMU/GNSS integration Kalman filter were the UAV navigation parameter error estimation, $\delta \hat{X}$ [21].

### 2.2. Error-State EKF Model

Using the error-state IMU equations as the system propagation equation, the state vector includes the IMU attitude error, IMU velocity error, IMU position error, inertial sensor biases, and GNSS receiver clock offset and drift for the error-state EKF [22]. The EKF is described as a system propagation equation (Equation (1)) and a measurement update equation (Equation (2)) [6]. In the measurement updates step, the GNSS pseudo-range and pseudo-range rate measurements are used for the measurement update, and the difference between the GNSS pseudo-range, GNSS pseudo-range rate and the corresponding predicted IMU pseudo-range, IMU pseudo-range rate constitute the EKF innovation. There is another expression of the system propagation equation, the full-state inertial navigation equations based EKF (i.e., the full-state EKF). For either the full-state or error-state in EKF, the essence is to estimate the state error. However, there is a difference between the error-state EKF and the full-state EKF in the system propagation step, the control input is added to the full-state EKF (Appendix A):(1)$${\hat{X}}_{k}^{-}=\Phi_{k}{\hat{X}}_{k-1}+\Gamma_{wk}{\vec{w}}_{k}$$
(2)$${\hat{X}}_{k}={\hat{X}}_{k}^{-}+K_{k}\underset{r_{k}}{\underset{︸}{\left({\vec{Z}}_{k}-H_{k}{\hat{X}}_{k}^{-}\right)}}$$
where ${\hat{X}}_{k}^{-}$ is the predicted error-state vector at epoch k, ${\hat{X}}_{k}$ is the measurement updated error-state vector at epoch k, *Φ_k_* is the dynamic state transition matrix at epoch k, *Γ_wk_* is the process noise matrix at epoch k, ${\vec{w}}_{k}$ is the process noise vector at epoch k, $K_{k}$ is the Kalman gain at epoch k ([App secB1-sensors-19-04912]), ${\vec{Z}}_{k}$ is the estimate of the measurement produced by the states estimated by the Kalman filter, $H_{k}$ is the measurement matrix at epoch k ([App secB1-sensors-19-04912]), and $r_{k}$ is the EKF innovation vector at epoch k. 

## 3. Error Overboundings Against IMU Faults

This part analyzes the propagation process of IMU faults for the error-state EKF in detail. Through the analysis, the recursive propagation equation between adjacent epochs of the EKF state error is obtained. Based on the EKF state error propagation equation and the EKF innovation equation, the error overboundings of the EKF state error are given.

### 3.1. IMU Faults Propagation in the Error-State EKF

In this section, the IMU faults propagation process in the state vector for the error-state EKF is explained. The differences between the output measurement value and the true value of the IMU are expressed in Equation (3):(3)$$\left\{ \begin{array}{c} \delta \varphi_{\beta }^{\alpha }={\tilde{\varphi }}_{\beta }^{\alpha }-\varphi_{\beta }^{\alpha }\\ \delta v_{\beta \alpha }^{r}={\tilde{v}}_{\beta \alpha }^{r}-v_{\beta \alpha }^{r}\\ \delta r_{\beta \alpha }^{r}={\tilde{r}}_{\beta \alpha }^{r}-r_{\beta \alpha }^{r} \end{array}\right.$$
where “~” represents the measurement value, $\varphi_{\beta }^{\alpha }$ is the attitude vector (rad), $v_{\beta \alpha }^{r}$ is the velocity vector (m/s), and $r_{\beta \alpha }^{r}$ is the position vector (m).

The inertial navigation error equations in the Earth-centered, Earth-fixed (ECEF) coordinate frame can be seen in Equation (4), including the attitude error, velocity error, and position error update equations:(4)$$\begin{array}{l} \delta {\dot{\vec{\varphi }}}_{eb}^{e}=C_{b}^{e}\left(\left[{\tilde{\vec{\omega }}}_{ib}^{b}\times \right]-\left[{\vec{\omega }}_{ib}^{b}\times \right]\right)-\left[{\vec{\omega }}_{ie}^{e}\times \right]\left({\tilde{\vec{\varphi }}}_{eb}^{e}-{\vec{\varphi }}_{eb}^{e}\right)\\ \delta {\dot{\vec{v}}}_{eb}^{e}=C_{b}^{e}\left({\tilde{\vec{f}}}_{ib}^{b}-{\vec{f}}_{ib}^{b}\right)-2\left[{\vec{\omega }}_{ie}^{e}\times \right]\left({\tilde{\vec{v}}}_{eb}^{e}-{\vec{v}}_{eb}^{e}\right)+{\hat{\vec{g}}}^{e}-{\vec{g}}^{e}\\ \delta {\dot{\vec{r}}}_{eb}^{e}={\tilde{\vec{v}}}_{eb}^{e}-{\vec{v}}_{eb}^{e} \end{array}$$
where $C_{b}^{e}$ is the coordinate transformation matrix from the body frame to the Earth frame (a detailed expression can be found in [App secB2-sensors-19-04912]), ${\vec{\omega }}_{ie}^{e}$ is the Earth rotation rate (rad/s), ${\vec{g}}^{e}$ is the gravity vector in the Earth-centered, Earth-fixed frame (m/s^2^), and [•×] represents the calculation of a skew-symmetric matrix.

Neglecting the effects of gravity errors, Equation (4) can be simplified as
(5)$$\begin{array}{l} \delta {\dot{\vec{\varphi }}}_{eb}^{e}=C_{b}^{e}\delta \left[{\vec{\omega }}_{ib}^{b}\times \right]-\left[{\vec{\omega }}_{ie}^{e}\times \right]\delta {\vec{\varphi }}_{eb}^{e}\\ \delta {\dot{\vec{v}}}_{eb}^{e}=C_{b}^{e}\delta {\vec{f}}_{ib}^{b}-2\left[{\vec{\omega }}_{ie}^{e}\times \right]\delta {\vec{v}}_{eb}^{e}\\ \delta {\dot{\vec{r}}}_{eb}^{e}=\delta {\vec{v}}_{eb}^{e} \end{array}.$$

Figure 2 shows the updating process of the attitude, velocity, and position in the ECEF coordinate frame for the IMU. When IMU faults occur in the gyroscope or accelerometer, the computed attitude, velocity, and position are sequentially affected, finally resulting in position errors.

Under the condition of a faulty IMU, the inertial navigation error equation in Equation (5) becomes Equation (6), where the apostrophe (’) indicates the faulty parameter. The parameters affected by the fault are expressed as the sum of a normal error term and a faulty additional error term, as shown in Equation (7):(6)$$\begin{array}{l} \delta {{\dot{\vec{\varphi }}}_{eb}^{e}}^{\prime }=C_{b}^{e}{\left(\delta \left[{\vec{\omega }}_{ib}^{b}\times \right]\right)}^{\prime }-\left[{\vec{\omega }}_{ie}^{e}\times \right]\delta {\vec{\varphi }}_{eb}^{e}\\ \delta {{\dot{\vec{v}}}_{eb}^{e}}^{\prime }={\left(C_{b}^{e}\right)}^{\prime }{\left(\delta {\vec{f}}_{ib}^{b}\right)}^{\prime }-2\left[{\vec{\omega }}_{ie}^{e}\times \right]\delta {\vec{v}}_{eb}^{e}\\ \delta {{\dot{\vec{r}}}_{eb}^{e}}^{\prime }={\left(\delta {\vec{v}}_{eb}^{e}\right)}^{\prime } \end{array}$$
(7)$$\begin{array}{c} \delta {{\vec{\omega }}_{ib}^{b}}^{\prime }=\delta {\vec{\omega }}_{ib}^{b}+\Delta {\vec{\omega }}_{ib}^{b}\\ \delta {{\vec{f}}_{ib}^{b}}^{\prime }=\delta {\vec{f}}_{ib}^{b}+\Delta {\vec{f}}_{ib}^{b}\\ \delta {{\vec{v}}_{eb}^{e}}^{\prime }=\delta {\vec{v}}_{eb}^{e}+\Delta {\vec{v}}_{eb}^{e} \end{array}.$$

Substituting Equation (7) into Equation (6), the inertial navigation error equation under the IMU faults condition can be shown as
(8)$$\begin{array}{l} \delta {{\dot{\vec{\varphi }}}_{eb}^{e}}^{\prime }=C_{b}^{e}\left(\delta \left[{\vec{\omega }}_{ib}^{b}\times \right]+\Delta {\vec{\omega }}_{ib}^{b}\right)-\left[{\vec{\omega }}_{ie}^{e}\times \right]\delta {\vec{\varphi }}_{eb}^{e}\\ \delta {{\dot{\vec{v}}}_{eb}^{e}}^{\prime }=\left(C_{b}^{e}+\Delta C_{b}^{e}\right)\left(\delta {\vec{f}}_{ib}^{b}+\Delta {\vec{f}}_{ib}^{b}\right)-2\left[{\vec{\omega }}_{ie}^{e}\times \right]\delta {\vec{v}}_{eb}^{e}\\ \delta {{\dot{\vec{r}}}_{eb}^{e}}^{\prime }=\delta {\vec{v}}_{eb}^{e}+\Delta {\vec{v}}_{eb}^{e} \end{array}.$$

Rearrange Equation (8),
(9)$$\begin{array}{l} \delta {{\dot{\vec{\varphi }}}_{eb}^{e}}^{\prime }=C_{b}^{e}\delta \left[{\vec{\omega }}_{ib}^{b}\times \right]-\left[{\vec{\omega }}_{ie}^{e}\times \right]\delta {\vec{\varphi }}_{eb}^{e}+\underset{{\vec{f}}_{\delta {\dot{\vec{\varphi }}}_{eb}^{e}}}{\underset{︸}{C_{b}^{e}\Delta {\vec{\omega }}_{ib}^{b}}}\\ \delta {{\dot{\vec{v}}}_{eb}^{e}}^{\prime }=C_{b}^{e}\delta {\vec{f}}_{ib}^{b}-2\left[{\vec{\omega }}_{ie}^{e}\times \right]\delta {\vec{v}}_{eb}^{e}+\underset{{\vec{f}}_{\delta {\dot{\vec{v}}}_{eb}^{e}}}{\underset{︸}{C_{b}^{e}\Delta {\vec{f}}_{ib}^{b}+\Delta C_{b}^{e}\left(\delta {\vec{f}}_{ib}^{b}+\Delta {\vec{f}}_{ib}^{b}\right)}}\\ \delta {{\dot{\vec{r}}}_{eb}^{e}}^{\prime }=\delta {\vec{v}}_{eb}^{e}+\underset{{\vec{f}}_{\delta {\dot{\vec{r}}}_{eb}^{e}}}{\underset{︸}{\Delta {\vec{v}}_{eb}^{e}}} \end{array}$$
where ${\vec{f}}_{\delta {\dot{\vec{\varphi }}}_{b}^{e}}$, ${\vec{f}}_{\delta {\dot{\vec{v}}}_{eb}^{e}}$, and ${\vec{f}}_{\delta {\dot{\vec{r}}}_{eb}^{e}}$ represent the resultant bias vectors for the attitude, velocity, and position error states due to the IMU faults. In the EKF, the accelerometer measurement of the specific force ${\vec{f}}_{ib,meas}^{b}$ is represented as Equation (10), which includes the true specific force, the accelerometer bias, and the process noise. In addition, the gyroscope measurement of the angular rate, ${\vec{\omega }}_{ib,meas}^{b}$, is represented as Equation (11), which includes the true angular rate, the gyroscope bias, a g-dependent bias related to the specific force, and process noise. The accelerometer bias and gyroscope bias are usually modeled as constant errors [23]: (10)$${\vec{f}}_{ib,meas}^{b}={\vec{f}}_{ib}^{b}+{\vec{b}}_{a}+{\vec{w}}_{a}$$
(11)$${\vec{\omega }}_{ib,meas}^{b}={\vec{\omega }}_{ib}^{b}+{\vec{b}}_{g}+G_{g}{\vec{f}}_{ib}^{b}+{\vec{w}}_{g}.$$

According to Equations (10) and (11), the accelerometer measurement of the specific force error and the gyroscope measurement of the angular rate error are derived from the accelerometer measurement of the specific force, ${\vec{f}}_{ib,meas}^{b}$, and the gyroscope measurement of the angular rate, ${\vec{\omega }}_{ib,meas}^{b}$, as shown in Equation (12):(12)$$\begin{array}{ll} \delta {\vec{\omega }}_{ib}^{b} & ={\vec{\omega }}_{ib,meas}^{b}-{\vec{\omega }}_{ib}^{b}\\ & ={\vec{b}}_{g}+G_{g}{\vec{f}}_{ib}^{b}+{\vec{w}}_{g}\\ \delta {\vec{f}}_{ib}^{b} & ={\vec{f}}_{ib,meas}^{b}-{\vec{f}}_{ib}^{b}\\ & ={\vec{b}}_{a}+{\vec{w}}_{a} \end{array}.$$

Substituting Equation (12) into Equation (9), the inertial navigation error equation under the IMU faults condition is
(13)$$\begin{array}{l} \delta {{\dot{\vec{\varphi }}}_{eb}^{e}}^{\prime }=C_{b}^{e}\left[\left({\vec{b}}_{g}+G_{g}{\vec{f}}_{ib}^{b}+{\vec{w}}_{g}\right)\times \right]-\left[{\vec{\omega }}_{ie}^{e}\times \right]\delta {\vec{\varphi }}_{eb}^{e}+{\vec{f}}_{\delta {\dot{\vec{\varphi }}}_{eb}^{e}}\\ \delta {{\dot{\vec{v}}}_{eb}^{e}}^{\prime }=C_{b}^{e}\left[\left({\vec{b}}_{a}+{\vec{w}}_{a}\right)\times \right]-2\left[{\vec{\omega }}_{ie}^{e}\times \right]\delta {\vec{v}}_{eb}^{e}+{\vec{f}}_{\delta {\dot{\vec{v}}}_{eb}^{e}}\\ \delta {{\dot{\vec{r}}}_{eb}^{e}}^{\prime }=\delta {\vec{v}}_{eb}^{e}+{\vec{f}}_{\delta {\dot{\vec{r}}}_{eb}^{e}} \end{array}.$$

Rearranging Equation (13), the inertial navigation error equation under the IMU faults condition (Equation (8)) becomes a continuous form of an EKF error-state equation as a function of EKF error states, process noise, and the state fault vector, as shown in Equation (14):
(14)$$\begin{array}{l} \delta {{\dot{\vec{\varphi }}}_{eb}^{e}}^{\prime }=\underset{F_{\delta {\vec{\varphi }}_{eb}^{e}}\delta {\vec{\varphi }}_{eb}^{e}}{\underset{︸}{-\left[{\vec{\omega }}_{ie}^{e}\times \right]\delta {\vec{\varphi }}_{eb}^{e}+C_{b}^{e}\left[{\vec{b}}_{g}\times \right]}}+\underset{G_{g}{\vec{w}}_{g}}{\underset{︸}{C_{b}^{e}\left[\left(G_{g}{\vec{f}}_{ib}^{b}+{\vec{w}}_{g}\right)\times \right]}}+{\vec{f}}_{\delta {\dot{\vec{\varphi }}}_{eb}^{e}}\\ \delta {{\dot{\vec{v}}}_{eb}^{e}}^{\prime }=\underset{F_{\delta {\vec{v}}_{eb}^{e}}\delta {\vec{v}}_{eb}^{e}}{\underset{︸}{-2\left[{\vec{\omega }}_{ie}^{e}\times \right]\delta {\vec{v}}_{eb}^{e}+C_{b}^{e}\left[b_{a}\times \right]}}+\underset{G_{a}{\vec{w}}_{a}}{\underset{︸}{C_{b}^{e}\left[{\vec{w}}_{a}\times \right]}}+{\vec{f}}_{\delta {\dot{\vec{v}}}_{eb}^{e}}\\ \delta {{\dot{\vec{r}}}_{eb}^{e}}^{\prime }=\delta {\vec{v}}_{eb}^{e}+{\vec{f}}_{\delta {\dot{\vec{r}}}_{eb}^{e}} \end{array}.$$

Based on the integral relationship between the velocity and the position in the error-state inertial navigation equations, Equation (14) can be rewritten as
(15)$$\begin{array}{l} \delta {{\dot{\vec{\varphi }}}_{eb}^{e}}^{\prime }=F_{\delta {\vec{\varphi }}_{eb}^{e}}\delta {\vec{\varphi }}_{eb}^{e}+G_{g}{\vec{w}}_{g}+{\vec{f}}_{\delta {\dot{\vec{\varphi }}}_{eb}^{e}}\\ \delta {{\dot{\vec{v}}}_{eb}^{e}}^{\prime }=F_{\delta {\vec{v}}_{eb}^{e}}\delta {\vec{v}}_{eb}^{e}+G_{a}{\vec{w}}_{a}+{\vec{f}}_{\delta {\dot{\vec{v}}}_{eb}^{e}}\\ \delta {{\dot{\vec{r}}}_{eb}^{e}}^{\prime }=F_{\delta {\vec{r}}_{eb}^{e}}\delta {\vec{r}}_{eb}^{e}+G_{r}{\vec{w}}_{r}+{\vec{f}}_{\delta {\dot{\vec{r}}}_{eb}^{e}} \end{array}.$$

Hence, an EKF state equation under the IMU faults condition generalized as Equation (16) in a continuous form. The error-state vector contains the IMU position error, the IMU velocity error, the IMU attitude error, accelerometer biases, gyroscope biases, the GNSS receiver offset, and the GNSS receiver drift:(16)$${\vec{X}}^{\prime }=F\vec{X}+G_{w}\vec{w}+{\vec{f}}_{\dot{\vec{X}}}$$
where
$$\vec{X}=\left[\begin{array}{c} \begin{array}{c} \delta {\vec{\varphi }}_{eb}^{e}\\ \delta {\vec{v}}_{eb}^{e} \end{array}\\ \delta {\vec{r}}_{eb}^{b}\\ {\vec{b}}_{a}\\ \begin{array}{c} {\vec{b}}_{g}\\ \delta \rho_{c}\\ \delta {\dot{\rho }}_{c} \end{array} \end{array}\right],\vec{w}=\left[\begin{array}{c} {\vec{w}}_{a}\\ {\vec{w}}_{g} \end{array}\right],f_{\dot{\vec{X}}}=\left[\begin{array}{c} \begin{array}{c} {\vec{f}}_{\delta {\dot{\vec{\varphi }}}_{eb}^{e}}\\ {\vec{f}}_{\delta {\dot{\vec{v}}}_{eb}^{e}}\\ {\vec{f}}_{\delta {\dot{\vec{r}}}_{eb}^{e}}\\ 0_{3} \end{array}\\ \begin{array}{c} 0_{3}\\ 0\\ 0 \end{array} \end{array}\right],G_{w}=\left[\begin{array}{cc} 0_{3} & -C_{b}^{e}\\ -C_{b}^{e} & 0_{3}\\ 0_{3} & 0_{3}\\ 0_{3} & 0_{3}\\ 0_{3} & 0_{3}\\ 0 & 0\\ 0 & 0 \end{array}\right]$$
$$F=\left[\begin{array}{ccccccc} \left[{\vec{\omega }}_{ie}^{e}\times \right] & 0_{3} & 0_{3} & 0_{3} & C_{b}^{e} & 0 & 0\\ -\left[\left({\hat{C}}_{b}^{e}{\hat{f}}_{ib}^{b}\right)\times \right] & -2\left[{\vec{\omega }}_{ie}^{e}\times \right] & 0 & -C_{b}^{e} & 0_{3} & 0 & 0\\ 0_{3} & I_{3} & 0 & 0_{3} & 0_{3} & 0 & 0\\ 0_{3} & 0_{3} & 0_{3} & 0_{3} & 0_{3} & 0 & 0\\ 0_{3} & 0_{3} & 0_{3} & 0_{3} & 0_{3} & 0 & 0\\ 0 & 0 & 0 & 0 & 0 & 0 & I\\ 0 & 0 & 0 & 0 & 0 & 0 & 0 \end{array}\right].$$

The discrete form of Equation (16) is shown in Equation (17):(17)$${{\vec{X}}_{k}}^{\prime }=\Phi_{k}{\vec{X}}_{k-1}+{\vec{\omega }}_{k}+{\vec{f}}_{{\vec{X}}_{k}}.$$

The discrete form of F is shown as
(18)$$\Phi \approx \left[\begin{array}{ccccccc} I_{3}-\left[{\vec{\omega }}_{ie}^{e}\times \right]\tau_{s} & 0_{3} & 0_{3} & 0_{3} & C_{b}^{e}\tau_{s} & 0 & 0\\ -\left[\left({\hat{C}}_{b}^{e}{\hat{f}}_{ib}^{b}\right)\times \right]\tau_{s} & I_{3}-2\left[{\vec{\omega }}_{ie}^{e}\times \right]\tau_{s} & {\left[{\vec{\omega }}_{ie}^{e}\times \right]}^{2}\tau_{s} & -C_{b}^{e}\tau_{s} & 0_{3} & 0 & 0\\ 0_{3} & I_{3}\tau_{s}-{\left[{\vec{\omega }}_{ie}^{e}\times \right]}^{2}\tau_{s}{}^{2} & I_{3}\tau_{s}+\left[{\vec{\omega }}_{ie}^{e}\times \right]\tau_{s}{}^{2}/2 & 0_{3} & 0_{3} & 0 & 0\\ 0_{3} & 0_{3} & 0_{3} & I_{3} & 0_{3} & 0 & 0\\ 0_{3} & 0_{3} & 0_{3} & 0_{3} & I_{3} & 0 & 0\\ 0 & 0 & 0 & 0 & 0 & I_{3} & \tau_{s}\\ 0 & 0 & 0 & 0 & 0 & 0 & I_{3} \end{array}\right]$$
where $\tau_{s}$ is the time interval of the adjacent propagation.

Thus, the predicted state under the IMU faults condition at epoch k is
(19)$${{\hat{X}}_{k}^{-}}^{\prime }=\Phi_{k}{\hat{X}}_{k-1}+{\vec{f}}_{{\vec{X}}_{k}}.$$

Substituting Equation (19) into Equation (2), the measurement update state under the IMU faults condition at epoch k is
(20)$$\begin{array}{ll} {{\hat{X}}_{k}}^{\prime } & ={{\hat{X}}_{k}^{-}}^{\prime }+{K_{k}}^{\prime }\left({\vec{Z}}_{k}-H_{k}{{\hat{X}}_{k}^{-}}^{\prime }\right)\\ & =\Phi_{k}{\hat{X}}_{k-1}+{\vec{f}}_{{\vec{x}}_{k}}+{K_{k}}^{\prime }\left({\vec{Z}}_{k}-H_{k}\left(\Phi_{k}{\hat{X}}_{k-1}+{\vec{f}}_{{\vec{X}}_{k}}\right)\right)\\ & =\left(I-{K_{k}}^{\prime }H_{k}\right)\left(\Phi_{k}{\hat{X}}_{k-1}+{\vec{f}}_{{\vec{X}}_{k}}\right)+{K_{k}}^{\prime }{\vec{Z}}_{k} \end{array}$$
where ${K_{k}}^{\prime }$ is the Kalman gain matrix computed by the faulty state transition matrix due to the IMU faults measurements and $H_{k}$ is the measurement matrix at epoch k ([App secB1-sensors-19-04912]).

To simplify Equation (20), we introduce a matrix ${L_{k}}^{\prime }$ as shown in Equation (21):(21)$${L_{k}}^{\prime }=\left(I-{K_{k}}^{\prime }H_{k}\right).$$
Subsequently, Equation (20) can be simplified as
(22)$${{\hat{X}}_{k}}^{\prime }={L_{k}}^{\prime }\Phi_{k}{\hat{X}}_{k-1}+{L_{k}}^{\prime }{\vec{f}}_{{\vec{x}}_{k}}+{K_{k}}^{\prime }{\vec{Z}}_{k}.$$
According to Equation (22), the additional term due to the IMU faults measurement at epoch k is ${L_{k}}^{\prime }{\vec{f}}_{{\vec{X}}_{k}}$.

### 3.2. EKF State Error Caused by IMU Faults

The PL against an IMU gyroscope and accelerometer fault should be formulated to overbound the state error with the integrity risk requirement. The true state is defined as shown in Equation (23):(23)$${\vec{X}}_{k}=\Phi_{k}{\vec{X}}_{k-1}+{\vec{\omega }}_{k}.$$

The state error propagation between two adjacent epochs can be derived as follows:

In step 1, multiply the matrix ${L_{k}}^{\prime }$ by the true state, ${\vec{X}}_{k}$:(24)$$\begin{array}{ll} {L_{k}}^{\prime }{\vec{X}}_{k} & ={L_{k}}^{\prime }\left(\Phi_{k}{\vec{X}}_{k-1}+{\vec{\omega }}_{k}\right)\\ & ={L_{k}}^{\prime }\Phi_{k}{\vec{X}}_{k-1}+{L_{k}}^{\prime }{\vec{\omega }}_{k} \end{array}.$$

In step 2, subtract ${L_{k}}^{\prime }{\vec{X}}_{k}$ from ${{\hat{X}}_{k}}^{\prime }$ in Equation (22):(25)$${{\hat{X}}_{k}}^{\prime }-{L_{k}}^{\prime }{\vec{X}}_{k}={L_{k}}^{\prime }\Phi_{k}{\hat{X}}_{k-1}+{L_{k}}^{\prime }{\vec{f}}_{{\vec{x}}_{k}}+{K_{k}}^{\prime }{\vec{Z}}_{k}-{L_{k}}^{\prime }{\vec{X}}_{k}.$$

In step 3, substitute ${\vec{Z}}_{k}=H_{k}{\vec{X}}_{k}+{\vec{v}}_{k}$ into Equation (25):(26)$$\begin{array}{ll} {{\hat{X}}_{k}}^{\prime }-{L_{k}}^{\prime }{\vec{X}}_{k} & ={L_{k}}^{\prime }\Phi_{k}{\hat{X}}_{k-1}+{L_{k}}^{\prime }{\vec{f}}_{{\vec{X}}_{k}}+{K_{k}}^{\prime }\left(H_{k}{\vec{X}}_{k}+{\vec{v}}_{k}\right)-{L_{k}}^{\prime }\left(\Phi_{k}{\vec{X}}_{k-1}+{\vec{\omega }}_{k}\right)\\ & ={L_{k}}^{\prime }\Phi_{k}\underset{\delta X_{k-1}}{\underset{⎵}{\left({\hat{X}}_{k-1}-{\vec{X}}_{k-1}\right)}}+{L_{k}}^{\prime }{\vec{f}}_{{\vec{X}}_{k}}+{K_{k}}^{\prime }{\vec{v}}_{k}-{L_{k}}^{\prime }{\vec{\omega }}_{k}+\underset{\leftarrow }{\underset{⎵}{{K_{k}}^{\prime }H_{k}{\vec{X}}_{k}}} \end{array}.$$

Substitute Equation (21) into Equation (26):(27)$$\begin{array}{l} \underset{\delta X_{k}}{\underset{⎵}{{{\hat{X}}_{k}}^{\prime }-{\vec{X}}_{k}}}={L_{k}}^{\prime }\Phi_{k}\underset{\delta x_{k-1}}{\underset{⎵}{\left({\hat{X}}_{k-1}-{\vec{X}}_{k-1}\right)}}+{L_{k}}^{\prime }{\vec{f}}_{{\vec{X}}_{k}}+{K_{k}}^{\prime }{\vec{v}}_{k}-{L_{k}}^{\prime }{\vec{\omega }}_{k}\\ \;\;\;\;\;↓\\ \;\;\delta X_{k}={L_{k}}^{\prime }\Phi_{k}\delta X_{k-1}+{L_{k}}^{\prime }{\vec{f}}_{{\vec{X}}_{k}}+{K_{k}}^{\prime }{\vec{v}}_{k}-{L_{k}}^{\prime }{\vec{\omega }}_{k} \end{array}.$$

As shown in Equation (27), the state error vector at epoch k can be expressed as a function of the previous state error vector, a state fault vector, a measurement noise vector, and a process noise vector, indicating that the derived state error vector affected by the IMU faults is recursive.

### 3.3. Error Overboundings

In the tightly coupled IMU/GNSS integration architecture, the EKF innovation is defined as the pseudo-range and the pseudo-range rate differences between the GNSS measurements and IMU predictions. The pseudo-range and pseudo-range rate measurements are obtained from the GNSS ranging processor. The pseudo-range and pseudo-range rate predictions are constructed using the corrected inertial navigation parameters, estimated receiver clock offset and clock drift, and satellite positions and velocities calculated according to the ephemeris. The relationship between the innovation and the state error under the IMU faults condition can be derived from the innovation definition as described below.

For the error-state EKF, the innovation at epoch k can be expressed as follows:(28)$$\begin{array}{ll} {\vec{\gamma }}_{k} & ={\vec{Z}}_{k}-H_{k}{{\hat{X}}_{k}^{-}}^{\prime }\\ & =H_{k}{\vec{X}}_{k}+{\vec{v}}_{k}-H_{k}{{\hat{X}}_{k}^{-}}^{\prime }\\ & =H_{k}\left(\Phi_{k}{\vec{X}}_{k-1}+{\vec{\omega }}_{k}\right)+{\vec{v}}_{k}-H_{k}\left(\Phi_{k}{\hat{X}}_{k-1}+{\vec{f}}_{{\vec{X}}_{k}}\right)\\ & =H_{k}\Phi_{k}\underset{-\delta X_{k-1}}{\underset{⎵}{\left({\vec{X}}_{k-1}-{\hat{X}}_{k-1}\right)}}+H_{k}{\vec{\omega }}_{k}+{\vec{v}}_{k}-H_{k}{\vec{f}}_{{\vec{X}}_{k}}\\ & =-H_{k}\Phi_{k}\delta X_{k-1}+H_{k}{\vec{\omega }}_{k}+{\vec{v}}_{k}-H_{k}{\vec{f}}_{{\vec{X}}_{k}} \end{array}.$$
Due to the relationship between the innovation and the state error, the error overboundings for the error-state EKF under the IMU faults can be expressed by the innovation. To simplify Equation (28), we introduce a matrix $U_{k}$ as shown in Equation (29):(29)$$U_{k}=\Phi_{k}\delta X_{k-1}+{\vec{f}}_{{\vec{X}}_{k}}$$
which is a common term in both the state error, in Equation (27), and the EKF innovation, in Equation (28), representing the impact caused by the IMU faults. Substitute Equation (29) into Equations (27) and (28):(30)$$\delta X_{k}={L_{k}}^{\prime }U_{k}+{K_{k}}^{\prime }{\vec{v}}_{k}-{L_{k}}^{\prime }{\vec{\omega }}_{k}$$
(31)$${\vec{\gamma }}_{k}=-H_{k}U_{k}+H_{k}{\vec{\omega }}_{k}+{\vec{v}}_{k}.$$

As an EKF does not take measurements for all the EKF states in the IMU/GNSS integration architecture, $\left(H_{k}{}^{T}H_{k}\right)$ becomes a singular matrix and is not an invertible matrix. Thus, as shown in Equation (32), we define $U_{k,p}$ as a partial matrix of $U_{k}$, only computed by the nonzero terms of $H_{k}$. The state error can be captured from the EKF innovation vector:(32)$$\begin{array}{l} -H_{k}U_{k}={\vec{\gamma }}_{k}-H_{k}{\vec{\omega }}_{k}-{\vec{v}}_{k}\\ \;\;\;\;↓\\ \;U_{k,p}=-{\left(H_{k,p}{}^{T}H_{k,p}\right)}^{-1}H_{k,p}{}^{T}\left({\vec{\gamma }}_{k}-H_{k}{\vec{\omega }}_{k}-{\vec{v}}_{k}\right) \end{array}.$$

Substituting Equation (32) into Equation (30), the state error can be expressed by using the EKF innovation vector, which can be accessed in real time as shown in Equation (33):(33)$$\begin{array}{ll} \delta X_{k,p} & ={L_{k,p}}^{\prime }U_{k,p}+{K_{k,p}}^{\prime }{\vec{v}}_{k}-{L_{k,p}}^{\prime }{\vec{\omega }}_{k,p}\\ & ={L_{k,p}}^{\prime }\left(-{\left(H_{k,p}{}^{T}H_{k,p}\right)}^{-1}H_{k,p}{}^{T}\left({\vec{\gamma }}_{k}-H_{k}{\vec{\omega }}_{k}-{\vec{v}}_{k}\right)\right)+{K_{k,p}}^{\prime }{\vec{v}}_{k}-{L_{k,p}}^{\prime }{\vec{\omega }}_{k,p}\\ & =-{L_{k,p}}^{\prime }{\left(H_{k,p}{}^{T}H_{k,p}\right)}^{-1}H_{k,p}{}^{T}{\vec{\gamma }}_{k}+{L_{k,p}}^{\prime }{\left(H_{k,p}{}^{T}H_{k,p}\right)}^{-1}H_{k,p}{}^{T}{\vec{v}}_{k}+{K_{k,p}}^{\prime }{\vec{v}}_{k} \end{array}.$$

To simplify Equation (33), introduce a matrix $V_{k}$ as shown in Equation (34):(34)$$V_{k,p}={L_{k,p}}^{\prime }{\left(H_{k,p}{}^{T}H_{k,p}\right)}^{-1}H_{k,p}{}^{T}.$$

Thus, the state error can be expressed as the sum of the innovation and the measurement noise by substituting Equation (34) into Equation (33):(35)$$\delta X_{k,p}=-V_{k,p}{\vec{\gamma }}_{k}+V_{k,p}{\vec{v}}_{k}+{K_{k,p}}^{\prime }{\vec{v}}_{k}.$$

According to Equation (35), the state error under the IMU faults condition can be bounded based on the distribution of the measurement noise. With the assumption of the measurement noise obeying a normal distribution, the error overboundings for the EKF state can be expressed as

(36)$$PL_{error-overboundings}=-V_{k,p}{\vec{\gamma }}_{k}\pm K_{md,IMU}\left\{\sigma_{V_{k,p}{\vec{v}}_{k},v}+\sigma_{{K_{k,p}}^{\prime }{\vec{v}}_{k},v}\right\}.$$

In Equation (36), the uncertainty noise bounding terms include the bias estimates term, $\sigma_{V_{k,p}{\vec{v}}_{k},v}$, the remaining position error term, $\sigma_{{K_{k,p}}^{\prime }{\vec{v}}_{k},v}$, and the coefficient, $K_{md,IMU}$, which is related to the integrity risk requirement allocated to the IMU faults condition of the IMU/GNSS integrated navigation architecture. Due to the fact that the IMU sensor bias estimates are computed in real time, it is highly probable that the error overboundings can overbound the state error for two sides [19]. This leads to a lower level of conservatism than using the absolute error overboundings.

## 4. Simulation and Analysis

### 4.1. Simulation Conditions

Simulations were conducted to show the error overboundings for UAVs under the IMU faults condition using both the full-state EKF (Appendix C) and the error-state EKF. In the simulations, the GNSS measurement noise and the IMU gyroscope and accelerometer measurement noise were modeled as Gaussian white noise, and the corresponding noise parameters were also used in the error overboundings.

The specific UAV trajectory can be seen in Figure 3, configured as an integration of two 45° turns and one vertical climb. The simulation time was 7 min.

The IMU sensors were set as consumer-grade inertial sensors with the noise parameters listed in Table 1 [24]. In the simulation, a 24-satellite GPS constellation was used, and the mask angle was 7.5°. The GNSS measurement noise was introduced into the pseudo-ranges as white Gaussian noise with a mean of zero and a variance of 30 m^2^. The data update frequency was 5 Hz for the GNSS receiver and 100 Hz for the IMU sensors. The integrated Kalman filter update was only performed when the GNSS signal was updated, otherwise only the IMU output was used. The output frequency was 100 Hz, which was the same as the IMU update frequency.

All simulations were performed using MATLAB R2018a on a Windows 10 PC with 12 GB of RAM and an Intel Core i7-6700 3.4 GHz processor.

### 4.2. Simulation Under Injected IMU Gyroscope and Accelerometer Faults

For the simulations of the error overboundings, the prior probabilities of an IMU gyroscope and accelerometer sensor faults were assumed to be 10^−3^, considering the IMU hardware performance used for a low-cost use and consumer-grade inertial device [25]. In the simulation, the ramp IMU gyroscope and accelerometer faults, with magnitudes of 1.5 m/s^2^ in the accelerometer and 0.03 rad/s in the gyroscope, was injected into the IMU accelerometer and gyroscope for three axes after 200 s. The faults injection lasted 10 s.

As an example, the vertical position errors are presented as the red dotted curve, which is the difference between the EKF estimated vertical position and the true vertical position, in Figure 4. In addition, the vertical error overboundings (VPL_error.overboundings_) are presented in Figure 4, as blue solid curves for the error-state EKF and green solid curves for the full-state EKF. In order to intuitively present the vertical state error in Figure 4, the state error is represented in Figure 5, as a blue dotted curve for the error-state EKF and a red solid curve for the full-state EKF. In this simulation, the value of K_md,IMU_ was chosen to be 3.29, which overbounds the noise of the state bias estimates with a missed detection probability of 10^−3^ ([App secB3-sensors-19-04912]). The simulation results show that the overboundings had the same trend as the position error and that the position error can be accurately overbounded during UAV operations under the missed detection probability requirement. Moreover, the error overboundings of the two EKFs were the same when no fault occurred. When faults were injected, the error overboundings for the error-state were higher than the full-state; the reason for this is that the error-state EKF is more sensitive to the faults than the full-state EKF. The east and north state errors and overboundings (i.e., east error overboundings (EPL_error.overboundings_), north error overboundings (NPL_error.overboundings_)) are also presented in Figure 6 and Figure 7, respectively, which further proves that the position error can be accurately overbounded and that the error overboundings of the two EKFs are the same in all directions.

### 4.3. Computational Efficiency Comparison

To compare the efficiency of the full-state and the error-state EKFs in the error overboundings calculation, the filtering time at each epoch is recorded in Figure 8. The filtering time of the error-state EKF was lower and more stable than that of the full-state EKF. The average filtering time for the error-state EKF was approximately 7.573 × 10^−4^ s, 16.163% less than that for the full-state error, approximately 9.033 × 10^−3^ s. A detailed comparison can be seen in Table 2.

Actually, the filtering time required for sparse matrix operations is usually proportional to the number of arithmetic operations in the number of nonzero elements. In the system state propagation period, the state transition matrix *Φ_k_* of the full-state EKF has more nonzero elements than that of the error-state EKF. As shown in Figure 8, taking, for example, the simulation times at 312.5 s and 357.5 s, there are distinct peaks for the full-state EKF because of the sparse matrices in the system propagation equation. Moreover, the convergence process of the EKF was evaluated by 3.66 times variance. As shown in Figure 9, the state error of the error-state EKF converged faster than the full-state EKF, the oscillation before convergence was smaller, and the convergence process was smoother.

## 5. Conclusions

The IMU faults propagation process was analyzed in this paper for the error-state EKF of tightly coupled IMU/GNSS integrated architecture. Accordingly, the error overboundings against the IMU faults was obtained. The error overboundings for the full-state and the error-state EKFs presented the same form in theoretical derivation, consisting of terms relating to the EKF innovation, the estimated bias, and the remaining position error. The simulation result shows that the derived error overboundings can fit the position error, reflecting the position error change caused by the IMU faults. Moreover, the error overboundings of the error-state EKF were the same as that of the full-state EKF in mathematical simulation. As the error-state EKF is more practical and its error overboundings calculation time was lower, the error-state EKF is recommended to assure the integrity of the IMU/GNSS integrated architecture for an UAV. The proposed error overbounding method with real collected data will be carried out in future work.

## Figures and Tables

**Figure 1 sensors-19-04912-f001:**
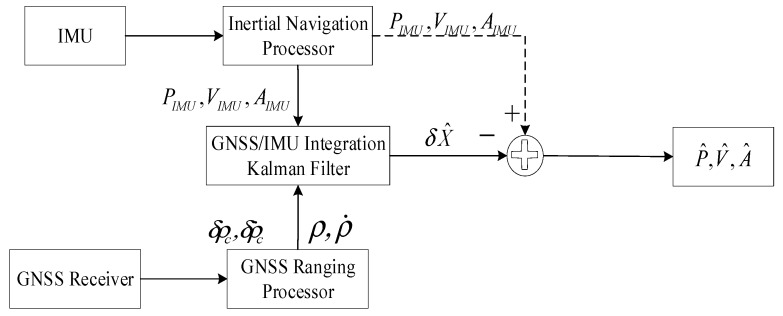
Tightly coupled inertial measurement unit (IMU)/global navigation satellite system (GNSS) integration architecture.

**Figure 2 sensors-19-04912-f002:**
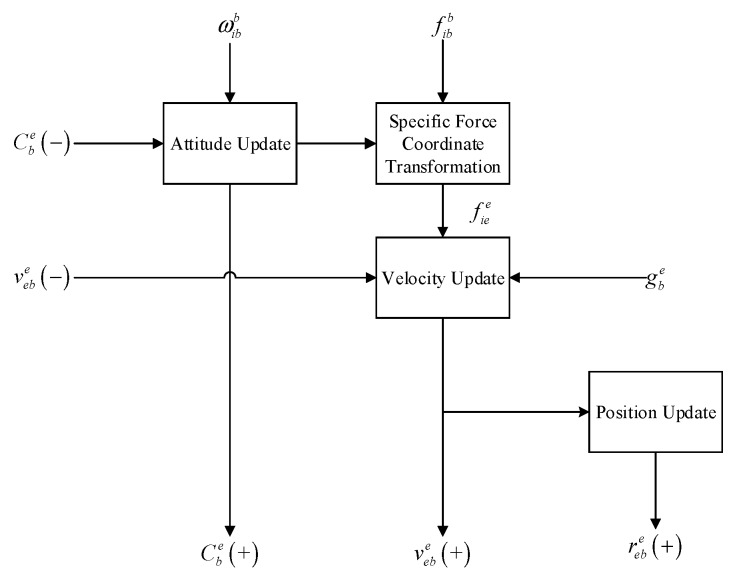
Navigation parameters update process for the IMU.

**Figure 3 sensors-19-04912-f003:**
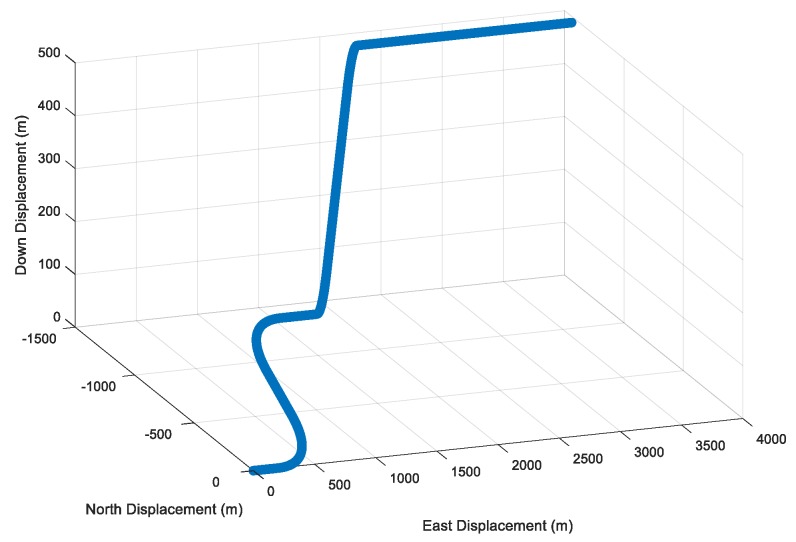
Unmanned aerial vehicle (UAV) trajectory in the simulation tests.

**Figure 4 sensors-19-04912-f004:**
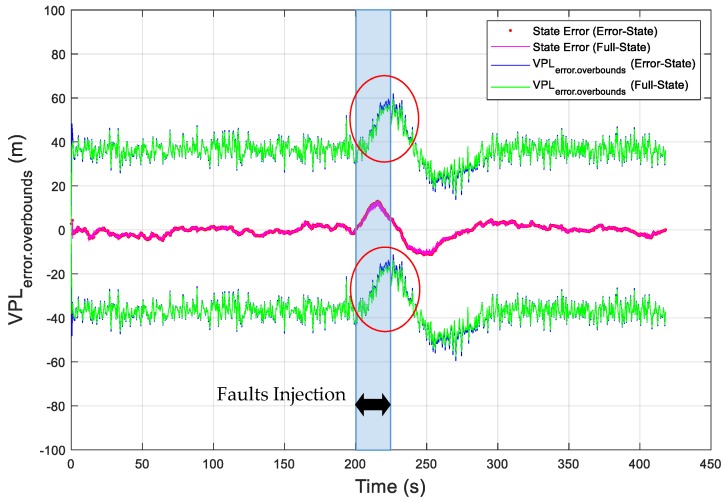
Simulation of the obtained error overboundings for the vertical position error.

**Figure 5 sensors-19-04912-f005:**
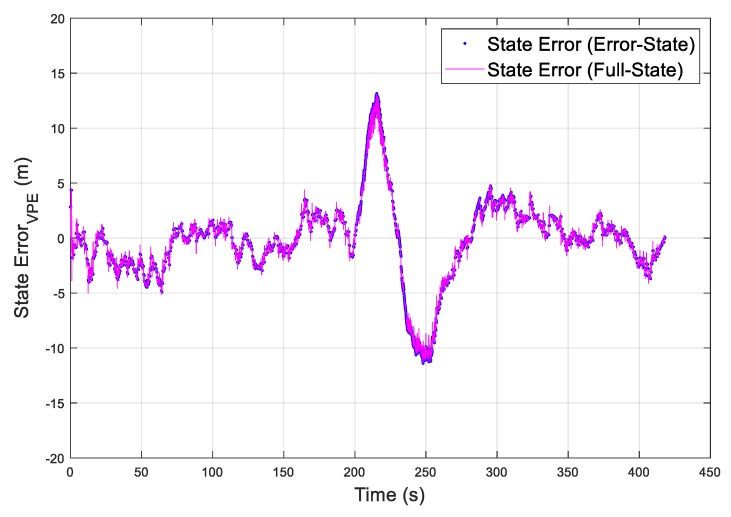
Simulation of the vertical position state error.

**Figure 6 sensors-19-04912-f006:**
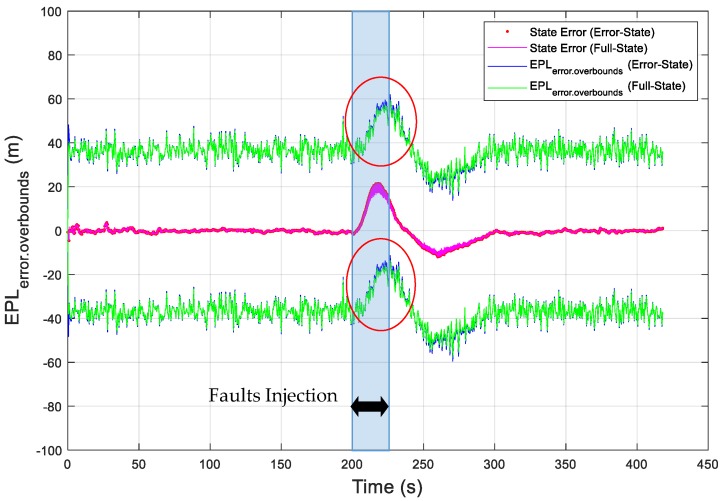
Simulation of the obtained error overboundings for the east position error.

**Figure 7 sensors-19-04912-f007:**
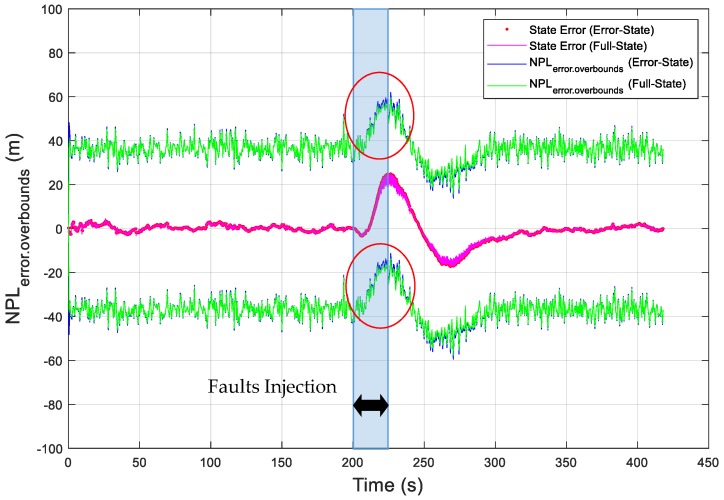
Simulation of the obtained error overboundings for the north position error.

**Figure 8 sensors-19-04912-f008:**
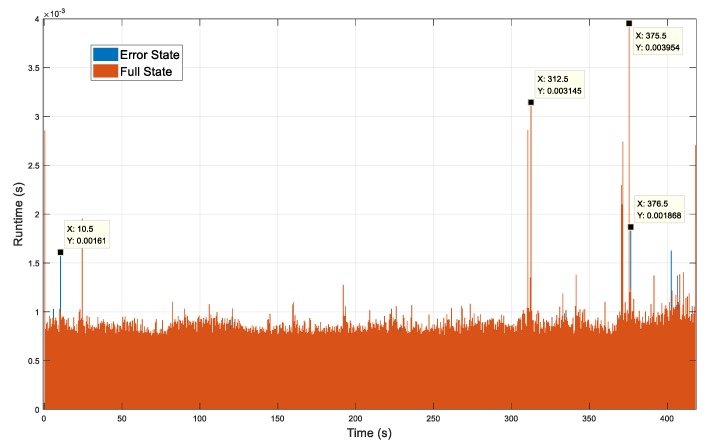
The filtering time of the error-state and full-state EKFs.

**Figure 9 sensors-19-04912-f009:**
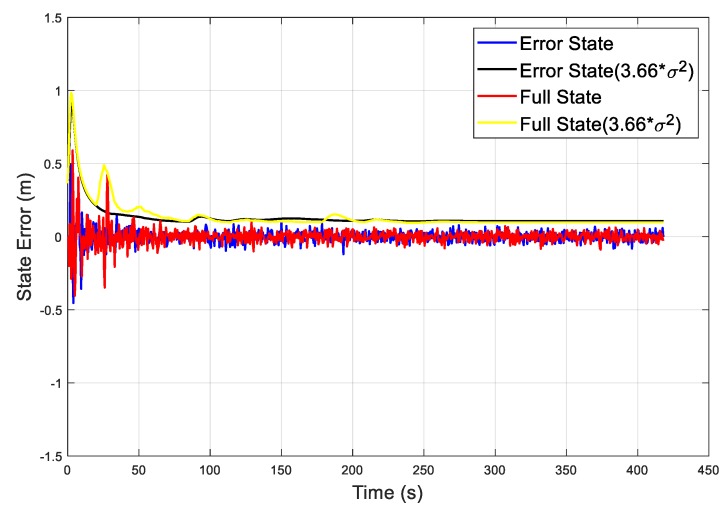
The state error of the error-state and full-state EKFs.

**Table 1 sensors-19-04912-t001:** IMU noise simulation parameters.

IMU Sensor (Consumer-Grade)			
Accelerometer		Gyroscope	
Noise ($\left[\mu g/\sqrt{Hz}\right]$)	Bias Noise (μg)	Noise ($\left[{}^{\circ}/h/\sqrt{Hz}\right]$)	Bias Noise ([°/h])
120	150	50	15

**Table 2 sensors-19-04912-t002:** Filtering time comparison.

Scheme	Maximum	Minimum	Mean	Variance
Error-state	1.868 × 10^−3^ s	8.03 × 10^−5^ s	7.573 × 10^−4^ s	4.8364 × 10^−8^
Full-state	3.954 × 10^−3^ s	9.21 × 10^−5^ s	9.033 × 10^−4^ s	1.4365 × 10^−7^

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
