# Peer review of "Comparative Analysis between Error-State and Full-State Error Estimation for KF-Based IMU/GNSS Integration against IMU Faults"

_sensors, 2019, doi:10.3390/s19224912_

Round 1

Reviewer 1 Report

The manuscript analyzes the IMU fault propagation process for an error-state EKF compared to a full-state EKF. From a methodological point of view, the paper appears properly structured and written. However, from an implementation’s perspective there are some critical points that need to be clarified:

It is not very clear how the Kalman gain matrix Kk' is being calculated. The matrix Hk for the EKF is the Jacobian matrix, which contains the partial derivatives of the observation functions. Could you please comment on that? Furthermore, nor system noise neither measurement covariance matrices are properly incorporated into the modelling process. White Gaussian noise is considered but it is not clear on which basis this is chosen. Why not consider specific devices/sensors with specific characteristics? This will give an added value on the applicability of the proposed methodology. I would say that it is very convenient to choose the same update frequency for both GNSS and IMU. However, usually this is not the case and the GNSS receiver has a lower update frequency compared to the IMU. This should be examined. I would recommend a combination of 20Hz for the GNSS and 200 HZ (or a similar one) for the IMU or a similar difference.

Some more specific comments are the following:

Consider adding a nomenclature. It will definitely improve the overall readability. Line 104: As this figure is presented, the reader does not have a clear view of the state variables. I would recommend adding them in the text before Figure 1. Line 141: please provide the coordinate transformation matrix. Lines 285-286: Why 10-3? Line 294: Why 3.29? Figure 4, 5, and 6: The red dots are not visible. In fact, the reader is confused with the red oval shape. Maybe two graphs with the state error should be given in another graph with smaller ranges for the y-axis.

Reviewer 2 Report

The paper studies and derives the error overboundings of the UAV navigation through tightly coupled GNSS/IMU integration error states EKF to analyze the IMU faults propagation process. It compares between the error overboundings using full state EKF and the error states EKF techniques.

COMMENTS:

Line 35: Redundant IMU could be mounted on small UAV as MEMS based IMU is very small size, low-cost, and has a moderate power consumption.

Line 104 (figure1): the output of the Inertial Navigation Processor is P, V, and A. However, this should be in more details as the Inertial Navigation Processor calculates the IMU pseudorange and IMU pseudorange rate. Then, In the figure this step should be clearly stated.

Line 111: It should be mentioned that the measurements update step could be implemented as error state EKF which is the difference between the GNSS pseudorange and predicted IMU pseudorange as well as the difference between the pseudorange rate of the GNSS and the predicted IMU.  

Line 188: the EKF error states are the position, velocity, attitude, accelerometers and gyroscopes biases only or the GNSS receiver clock offset and drift should be added in the error state EKF.

Line 247: The state error is much better to be used in this context instead of the position state error.

Line 279: In experimental results, the data rate is 100 Hz for both GNSS and IMU which is not practical for GNSS receiver to reach to 100 Hz in real experiments. From practical point of view, the data rate of the IMU is usually higher than that of the GNSS data rate.

Line 294: At what basis the K coefficient is chosen to be 3.29?

Line 334, and 338: Moreover, the error overboundings of the error-state EKF is the same as that of the full-state EKF. What is meant by this statement? Need more explanation.

Language Comments:

Line15: the estimated bias instead of the estimate bias.

Line72: the IMU faults propagate instead of the IMU faults propagates.

Line113: updates instead of update.

Line 146: attitude, velocity, and position instead of attitude, velocity and position

Line 147: When IMU faults occur instead of When IMU faults occurs.

Lines 163, 164: The accelerometer bias and the process noise instead of bias in the accelerometer and process noise.

Lines 208: The true state is defined as shown instead of We define the true state.

Line 242: all of the EKF states instead of all of the EKF state.

Lines 242 to 245: It should be re paraphrased.  

Line298: When there is no fault occurs instead of When there is no fault occur

Line 335: the estimated bias instead of the estimate bias.

Line 370: Extended Kalman Filter instead of extend Kalman filter.

Line 371: For nonlinear systems instead of For nonlinear system

Line 385: When IMU faults occur instead of When IMU faults occurs.

Line 388: parameters instead of parameter.

Line 400: for the attitude, velocity, and position states instead of for the attitude, velocity and position states.

Line 461: IMU faults present instead of IMU faults presents.

General Comments:

1- Could the experimental results be real rather than simulated? (collect real data and then induced some simulated IMU faults) or it can be simulated but the data rate of GNSS should be practical (not a 100 Hz)?

2- What are the types of the stochastic model used for accelerometers and gyroscopes biases?

Round 2

Reviewer 1 Report

I would like to thank the authors for revising their manuscript. It can now be accepted for publication.